# Metabolomics Provide Sensitive Insights into the Impacts of Low Level Environmental Contamination on Fish Health—A Pilot Study

**DOI:** 10.3390/metabo10010024

**Published:** 2020-01-06

**Authors:** Sara M. Long, Dedreia L. Tull, David P. De Souza, Konstantinos A. Kouremenos, Saravanan Dayalan, Malcolm J. McConville, Kathryn L. Hassell, Vincent J. Pettigrove, Marthe Monique Gagnon

**Affiliations:** 1Centre for Aquatic Pollution Identification and Management (CAPIM), Bio21 Molecular Science and Biotechnology Institute, The University of Melbourne, 30 Flemington Road, Parkville, VIC 3010, Australia; 2Aquatic Environmental Stress (AQUEST) Research Group, School of Science, RMIT University, PO Box 71, Bundoora, VIC 3083, Australia; kathryn.hassell@rmit.edu.au (K.L.H.); vincent.pettigrove@rmit.edu.au (V.J.P.); 3Metabolomics Australia, Bio21 Molecular Science and Biotechnology Institute, The University of Melbourne, 30 Flemington Road, Melbourne, VIC 3010, Australia; dedreia@unimelb.edu.au (D.L.T.); desouzad@unimelb.edu.au (D.P.D.S.); kkouremenos@trajanscimed.com (K.A.K.); saravanan.dayalan@gmail.com (S.D.); malcolmm@unimelb.edu.au (M.J.M.); 4Department of Biochemistry and Molecular Biology, Bio21 Molecular Science and Biotechnology Institute, The University of Melbourne, 30 Flemington Road, Parkville, VIC 3010, Australia; 5Centre for Aquatic Pollution Identification and Management (CAPIM), The University of Melbourne, Parkville, VIC 3010, Australia; 6School of Molecular and Life Sciences, Curtin University, Bentley, WA 6102, Australia; m.gagnon@curtin.edu.au

**Keywords:** biomarkers, fatty acid metabolites, energy metabolism, amino acids, metals, diet, flathead, *Platycephalus bassensis*

## Abstract

This exploratory study aims to investigate the health of sand flathead (*Platycephalus bassensis*) sampled from five sites in Port Phillip Bay, Australia using gas chromatography-mass spectrometry (GC-MS) metabolomics approaches. Three of the sites were the recipients of industrial, agricultural, and urban run-off and were considered urban sites, while the remaining two sites were remote from contaminant inputs, and hence classed as rural sites. Morphological parameters as well as polar and free fatty acid metabolites were used to investigate inter-site differences in fish health. Significant differences in liver somatic index (LSI) and metabolite abundance were observed between the urban and rural sites. Differences included higher LSI, an increased abundance of amino acids and energy metabolites, and reduced abundance of free fatty acids at the urban sites compared to the rural sites. These differences might be related to the additional energy requirements needed to cope with low-level contaminant exposure through energy demanding processes such as detoxification and antioxidant responses as well as differences in diet between the sites. In this study, we demonstrate that metabolomics approaches can offer a greater level of sensitivity compared to traditional parameters such as physiological parameters or biochemical markers of fish health, most of which showed no or little inter-site differences in the present study. Moreover, the metabolite responses are more informative than traditional biomarkers in terms of biological significance as disturbances in specific metabolic pathways can be identified.

## 1. Introduction

The detection of subtle changes in animal physiology associated with low level exposure to different pollutants can be used as an early indication of the impacts of contamination, while also providing an understanding of the physiological effects induced by the exposure. Metabolomics approaches provide a sensitive assessment of animal physiology and can offer new insights into an organism’s biochemistry to complement traditional biomarkers of health in ecotoxicological studies [1,2,3]. Metabolomics approaches have been widely used in mammalian ecotoxicology [4,5], and to assess fish health, for example, identifying indicators of immune suppression [6], sexual maturity [7], and nutritional status [8]. Changes in metabolite profiles have also been used in some laboratory-based fish exposures [9,10,11] and to investigate the effects of pollution on field-collected marine fish (e.g., [12,13,14]). It is important to note that other factors may affect metabolite abundance in fish collected from the field such as temperature and diet [15,16,17], which may limit the application of metabolomics to detect impacts of contaminant exposure under field conditions. Furthermore, season and geography may also have an impact on fish responses to their local environment; however, in the present study, all fish were sampled within the same five-day period and from locations within the same embayment so these factors were thought to be less likely to impact responses than temperature and diet. This paper describes a pilot study that investigates the feasibility of using metabolomics approaches to assess the impact of local environmental conditions on the health of a resident fish in Port Phillip Bay, Victoria, Australia.

Port Phillip Bay is a large semi-closed embayment in southern Australia that is surrounded by the city of Melbourne (4.8 million residents) at the northern end of the bay and the city of Greater Geelong in the southwestern end (Figure 1). These areas contain heavily urbanized regions as well as manufacturing plants, oil refineries, and Australia’s busiest shipping port (the Port of Melbourne). Along the western side of Port Phillip Bay is the Western Treatment Plant (Figure 1), which discharges large quantities of treated effluent into Port Phillip Bay.

The Bay has a narrow entrance that restricts water exchange and leads to flushing times that in some areas exceed one year [18], and therefore any contamination input into the Bay is likely to remain for extensive periods. In the 1980s, stormwater and industrial effluent inputs caused extensive contamination by heavy metals, organochlorine pesticides (OCs), polycyclic aromatic hydrocarbons (PAHs), and polychlorinated biphenyls (PCBs) [19]. Subsequent management strategies implemented to improve the health of the Bay appear to have been effective, since recent studies have demonstrated low levels of legacy pollutants (such as PCBs, OCs, and PAHs) in Port Phillip Bay sediments and resident fish when compared to other urban embayments [20,21]. Furthermore, a study assessing the contaminant status of Port Phillip Bay stated that toxicants were not a concern in Port Phillip Bay except in a few localised areas including Corio Bay (inputs predominantly through shipping and refinery operations), Hobsons Bay (major sources of contamination are from catchment run-off from the Yarra River), and more urbanised estuaries within the Bay such as Mordialloc, receiving contaminated waters from Mordialloc Creek [22]. Specifically, Mordialloc and Hobsons Bay had high concentrations of sediment-bound contaminants, mainly metals (namely zinc, copper, nickel, and mercury) and total petroleum hydrocarbons (Centre for Aquatic Pollution Identification and Management (CAPIM) unpublished data and [23]), that were above Australian environmental guideline values [24] and therefore a potential threat to the health of the local biota [22].

Sand flathead (*Platycephalus bassensis*) is a suitable sentinel species for ecotoxicological investigations in Port Phillip Bay as it is sedentary, non-migratory, and positioned high in the food chain [25,26] and has been used previously as an indicator of the pollution status of Port Phillip Bay [26,27] and in Tasmania [28,29]. Other findings have already been published on the same cohort of fish presented in this study, whereby tissue contaminant concentrations, a range of physiological indicators, and biomarkers of fish health demonstrate that sand flathead are still exposed to and exhibit the effects of low level contaminants in some parts of the Bay [21,30,31]. More specifically, whilst PAH, PCB, and OC concentrations were all below detection in edible muscle, some metals, in particular Hg, were detected (Table 1), albeit at levels below Australian Food Standards guideline values [21]. The sites where fish were found to have the highest concentrations of Hg in their tissues were the urbanised/industrialised sites of Mordialloc and Hobsons Bay. Similarly, using a suite of biomarkers including hepatic enzymes (Ethoxyresorufin O-Deethylase (EROD), carboxylesterase), biliary PAH metabolites, and DNA damage, Baker et al. [30] concluded that the health of fish collected from urban/industrialised areas (Corio Bay, Hobsons Bay, and Mordialloc) was compromised in comparison to fish sampled from less developed areas. Hepatic gene expression and liver histology analyses further support the notion that sand flathead from urban/industrialised areas within Port Phillip Bay exhibit indications of pollution stress, relative to fish from less developed/impacted areas [31].

The present study extends these analyses by using a combination of traditional parameters of fish health and advanced metabolite profiling technologies to assess sand flathead responses to low levels of urban contaminants. The study is exploratory in nature as very few metabolomics studies have been conducted on field-collected fish and as far as we are aware, this is the first study to do so with sand flathead. The aim of this study was to use a combination of metabolomics and morphological parameters of fish condition to assess the health of Port Phillip Bay sand flathead and to determine if metabolomics has the potential to discriminate between fish collected from rural (low contamination) sites and fish collected from sites that are the recipients of urban contamination.

## 2. Results

### 2.1. Fish Morphology 

All fish selected for metabolomics were females and determined to be two years old (based on otolith age estimation). Fish from Mordialloc were significantly shorter than fish from Corio Bay and Hobsons Bay (*p* < 0.05), but not different in length from fish from Sorrento and St. Leonards (Table 2). Fish from the sites at the rural, southern end of the Bay (i.e., Sorrento and St. Leonards) had significantly (*p* < 0.05) smaller livers relative to their body size (i.e., lower LSI), compared to fish from the urban sites of Corio Bay, Mordialloc, and Hobsons Bay at the more heavily urbanized areas of the Bay.

Larger liver mass relative to body mass may be suggestive of exposure to contamination [32]. It can also be influenced by reproductive status, since the liver plays a central role in the process of vitellogenesis [33,34]. Examination of the gonads revealed that three fish from Mordialloc and one from Hobsons Bay were at stages 2 or 3 (i.e., reproductively mature or not reproductively active), whereas all the other fish had gonads at stages 0 or 1 (reproductively immature [35]). This was not related to the differences in morphometric parameters between the sites, with all fish from Mordialloc and Hobsons Bay having a significantly greater LSI than fish from St. Leonards and Sorrento (*p* < 0.05). Therefore, our data suggest that the reproductive stage is unlikely to be the cause of the differences in the morphometric parameters observed in this study.

### 2.2. Fish Liver Metabolomics

#### 2.2.1. Polar Metabolites

Sixty-nine polar metabolites were identified in sand flathead livers from the GC-MS data including amino acids, organic acids, and metabolites involved in central carbon metabolism. To obtain an overview of the effects of site, a partial least squares discriminant analysis (PLS-DA) of the data matrix was carried out to investigate the overall behaviour of the polar metabolites between sites. There was a separation between Sorrento and St Leonards and three urban sites (Figure 2, cross validation results are presented in Appendix A). In addition, a principal component analysis (PCA) was also conducted, which showed separation between Sorrento and the other sites (data not shown).

Following multivariate analyses, a one-way ANOVA was conducted to determine if the metabolites differed significantly between sites. Sixteen metabolites were identified as being significantly different with an FDR adjusted *p* value < 0.05, and a further 11 ‘trending’ metabolites that had an FDR adjusted *p* value between 0.05 and 0.1 ([36], see Appendix A for a full list of significant metabolites and corresponding *p* and FDR-adjusted *p* values). These metabolites included amino acids (i.e., alanine, threonine, and tyrosine); intermediates in glycolysis (i.e., phosphoenol pyruvate and 3-phosphoglyceric acid), and the tricarboxylic acid (TCA) cycle (i.e., succinic acid).

Fishers LSD post hoc comparisons revealed differences in metabolite abundance between sites (Appendix A). The greatest number of differences in metabolites were observed in comparisons between fish from Sorrento and Mordialloc (16 metabolites), Sorrento and Corio Bay (13 metabolites), and Sorrento and St. Leonards (12 metabolites). The comparisons between the remaining sites identified a handful of metabolites that differed between the sites. Interestingly, most differences in metabolite abundance between fish from Sorrento and the other sites were amino acids and lactic acid (Appendix A); whereas variations between the other sites were a combination of different metabolite classes. In general, metabolites were present in lower abundance in fish sampled from the sites at the rural, southern end of the bay compared to the urbanised sites; for example, alanine, lysine, and lactate were in greater abundance in the livers of fish from the urban sites of Mordialloc, Corio Bay, and Hobsons Bay compared to Sorrento and St. Leonards (Figure 3).

Pathway analyses and over representation analyses (ORA) of the metabolites that were significantly different were carried out to determine which pathways were affected to better understand the biological significance of local site effects. These analyses revealed that the main pathways affected between the rural and urban sites included the production of amino acids (i.e., glycine, serine, and threonine metabolism) and energy metabolism (for example, gluconeogenesis, glycolysis, and the citric acid cycle) (Figure 3).

#### 2.2.2. Free Fatty Acid Metabolites

Twenty-four free fatty acids were detected in sand flathead livers including saturated and mono-, di-, and tri-unsaturated fatty acids. A PLS-DA showed the that there was little separation between the sites (Figure 4 and Appendix A show the cross validation results). An ANOVA identified two fatty acids that differed significantly between sites (false discovery rate (FDR) adjusted *p* value < 0.05) and a further three ‘trending’ metabolites (see Appendix A), which were mainly odd-chain saturated fatty acids (C15:0, C17:0, C21:0), docosanoic acid (C22:0), and the unsaturated fatty acid C18:3. In general, the abundance of free fatty acids in fish from Corio Bay were significantly lower than at the other sites. Flathead from Sorrento and St. Leonards had the highest median free fatty acid abundance, and Mordialloc and Hobsons Bay had levels that were intermediate (Figure 5). Fishers LSD post hoc-tests identified differences in metabolite abundance (Appendix A), with the greatest number of metabolite differences between fish from Corio Bay and Sorrento and St. Leonards. When comparing Corio Bay fish to Hobsons Bay and Mordialloc, there were only one and two metabolites, respectively, that were significantly different. There was no obvious pattern of differences between sites in terms of saturated and unsaturated fatty acids. There was a gradient in response to a higher abundance of free fatty acid metabolites at the rural sites and lower abundance at the more industrial and more densely populated sites of Corio Bay, Hobsons Bay, and Mordialloc (Figure 5 and Appendix A).

## 3. Discussion

### 3.1. Effects of Contamination

Sediments in creeks flowing into the Mordialloc and Hobsons Bay estuaries had high levels of metals and petroleum hydrocarbons compared to other parts of Port Phillip Bay [22], some of which exceeded national guideline values for the likelihood of toxic effects on biota [24]; in addition, Corio Bay had high levels of petroleum hydrocarbons [22]. Fish are known to take up and accumulate metals [37] and organic contaminants [26] from their local environment. In the present study, the fish were likely to have taken up contaminants via the diet (i.e., through consumption of contaminated prey [31]), as observed in other vertebrates in similar areas of Port Phillip Bay [38] and, to a lesser extent, from direct exposure to contaminated sediment. Therefore, the changes observed in the livers including higher LSI and differences in metabolite abundance are consistent with fish that have responded to contaminant exposure, as discussed below. Significant differences were observed in some metal concentrations in the white muscle of sand flathead between sites, for example, mercury was significantly higher at Mordialloc than at all the other sites [21]. The concentration of organic contaminants were below the limits of detection in all fish measured [21]. Indeed, it is known that PAHs and other organic contaminants are metabolised by fish [27,39] and eliminated via the bile [21,40].

The results from the biochemical pathway analyses suggest that pathways involved in energy metabolism were affected in sand flathead sampled at Mordialloc, Hobsons Bay, and Corio Bay. An increase in the abundance of amino acids was observed in fish livers from the urban sites compared to the rural sites, suggesting increased protein breakdown. Protein breakdown will result in the mobilisation of amino acids, which can be used in energy production to manage the additional energy demands for detoxification or antioxidant responses during contaminant exposure [41]. Alanine, lactic acid, and phosphoenol pyruvate are involved in gluconeogenesis and glycolysis [42], and in the present study, a significantly greater abundance of these metabolites was found in the livers of fish from the urban sites compared to those from the rural sites. These results further suggest that changes in energy metabolism were occurring.

Results from our study are consistent with findings from previous studies that found exposure to metals resulted in an increase in amino acids, changes in abundance of metabolites involved in energy metabolism, and an increase in aminotransferases (enzymes involved in protein catabolism) in the blood and livers of fish [41,43,44,45]. It was hypothesised that this would be related to an increased requirement for energy due to energy-demanding processes such as antioxidant defence mechanisms and metal-excreting/detoxification mechanisms (i.e., metallothionein production) [43]. Brandão et al. [12] also found changes in amino acids, lactic acid, and glucose abundance in the liver of golden grey mullet (*Liza aurata*) collected from a mercury-contaminated lagoon.

Previous studies have shown changes in the abundance of amino acids in the muscle tissue of fish exposed to crude oil and chemically dispersed crude oil [9,10] as well as intermediates in energy metabolism pathways such as succinate and lactate. This is also consistent with our results, however, there are fewer studies investigating the effects of petroleum hydrocarbons in fish using metabolomics than there are for metals, so comparisons are limited. A study investigating the effects of long term exposure to field-collected contaminated sediments (predominantly contaminated with PAHs) in flounder under laboratory conditions observed modest changes in the metabolite profiles between treatments, however, the authors suggested this may be due to low bioavailability of the PAHs [11]. This may also be the case in the present study in fish from Corio Bay (which has been historically exposed to petroleum hydrocarbon discharges [46]).

A decrease in the abundance of odd chain fatty acid metabolites was observed in livers of fish from Corio Bay and a moderate decrease in livers of fish from Mordialloc and Hobsons Bay compared to the rural sites. This is in contrast to results from a study by Speranza et al. [47] that found an increase in free fatty acids in livers of detritivorous fish (*Prochilodus lineatus*) collected in polluted areas compared to clean areas; however, this may also be related to the abundance of food at the polluted sites rather than pollution itself. Recent studies by Kowalczyk-Pecka et al. [48,49] have shown that fatty acid metabolites respond to a variety of xenobiotics in the land snail, *Helix pomatia* including metals and pesticides. The authors suggested that contaminants affected the physical and chemical properties of biological membranes and a reduction in free fatty acids was indicative of these alterations. In addition, fatty acids can be utilised as alternative sources of energy [49]. The decrease in the abundance of these metabolites in fish from the urban sites may suggest they are using alternative energy pathways to cope with contaminant exposure, however, the influence of diet on fatty acid metabolites must also be considered (see below).

### 3.2. Effect of Temperature and Diet

There are other environmental variables that may influence sand flathead metabolism and result in a change in the abundance of metabolites and morphological parameters between the sites including temperature and/or availability of food.

#### 3.2.1. Temperature

Although there were differences in temperature between the sites, the biggest difference in temperature at the time of sampling was between Sorrento and St Leonards (18.8 °C and 20.6 °C, respectively, Appendix A) and as their metabolite profile was relatively similar, it seems unlikely that temperature is responsible for the differences in metabolite abundance between the rural and urban sites.

#### 3.2.2. Diet

The composition of fatty acids in fish is influenced by several factors including habitat and foraging mode [50,51] and diet [52,53,54]. There were differences in primary productivity between the sites, with the northern sites of Hobsons Bay and Mordialloc having a higher concentration of phytoplankton than Corio Bay, Sorrento, and St Leonards (Appendix A), with Corio Bay having the lowest abundance overall. This difference in phytoplankton abundance is likely to have influenced the composition of local food webs and consequently, there will be differences in diet due to the availability of prey between sites. Phytoplankton provides the precursor molecules to produce odd chain fatty acids [55]. Hence, Lei et al. [50] linked the consumption of phytoplankton and bacteria to odd chain fatty acid abundance in fish livers. Although dietary analyses were not done with the fish used in this study, we propose that the significantly lower abundance of odd chain fatty acids in fish from Corio Bay may be related to the lower abundance of phytoplankton at that site, and the subsequent differences in food composition relative to the other sites. Speranza et al. [47] found that the abundance of lipids and free fatty acids in fish livers was related to the abundance of organic matter, which is similar to observations in the present study relating free fatty acid abundance with concentrations of phytoplankton present at the different sites. It is unclear if the abundance of phytoplankton is related to the low levels of petroleum hydrocarbons present at this site as this was beyond the scope of this study. This highlights the sensitivity of metabolomics approaches in unravelling the impacts of local conditions on ecological interactions and fish biology. Further research into the impacts of diet on metabolite profiles is required to understand the link between diet and free fatty acid profiles in sand flathead.

The fact that very few studies have, to date, used metabolomics approaches in wild-caught fish makes the present study exploratory in nature, and while additional sampling (e.g., seasonal or annual sampling as well as laboratory exposures) would be required to further correlate altered metabolite profiles and contaminant exposure and increasing the numbers of fish used for each treatment, the current results suggest the significant potential of metabolomics to detect the early impacts of contamination in field-caught organisms. Furthermore, impacts of other confounding factors such as diet/food availability, physiological state, and sex on fish metabolism need to be understood before metabolomics approaches can be routinely used as a biomonitoring tool.

### 3.3. Combining Traditional and Novel Biological Assessment Techniques in Understanding Fish Health

A handful of studies have combined the use of traditional indicators of fish health with metabolite biomarkers to assess the impacts of exposure to contamination under controlled laboratory conditions (for example, [2,45]). Traditional indicators of fish health include physiological measurements such as CF, LSI, or gonadosomatic index (GSI) as well as enzymatic markers such as ethoxyresorufin-O-deethylase (EROD) activity, stress proteins HSP70, or oxidative DNA damage to name a few. While these widely used indicators offer the advantage of low cost and ease of measurements, they also require chronic exposure to moderate to high contamination levels to be modulated. In addition, these parameters are often seen as isolated biomarkers of fish health, which, when considered individually, do not conclude in significant results [56]. On the other hand, untargeted and targeted metabolomics approaches integrate information on multiple metabolic pathways, which integrates both the genetics and physiological state of an organism as well as the interaction with the external environment [57]. For example, a recent study has highlighted the sensitivity of metabolomics approaches to detecting differences in populations of mussels (*Mytilus galloprovincialis*) due to differing local environmental conditions in a bay in south-eastern Australia [58]. Furthermore, if specific pathways or classes of metabolites are known to be important, targeted approaches can increase the sensitivity of the detection of these metabolites and allow for the detection of metabolic anomalies much earlier than traditional physiological or biochemical markers of fish health.

## 4. Materials and Methods 

### 4.1. Fish Collection and Processing

Southern sand flathead were collected by hook and line at five locations in Port Phillip Bay (Figure 1) between 25 February and 3 March 2015. All fish were handled as per conditions outlined in an approved animal ethics permit from Curtin University (AEC 2015_05), and collections were done under general research permit RP1216 (Department of Environment and Primary Industries, Victoria State Government). Sorrento and St. Leonards were classified as rural sites for this study; whereas Hobsons Bay, Mordialloc, and Corio Bay were classified as urban sites as per Sharp et al. [23] (see Appendix A for details of each site and Table 1 for contaminant concentrations in fish tissue) and Figure 1 for land uses in the surrounding areas.

Upon capture, fish were immediately placed in a tank on-board a dinghy with recirculating site water and transported to the Victorian Marine Sciences Consortium for dissection. All fish were processed within 2 h of capture and killed by iki jime [59]. Fish were individually examined for external parasites or anomalies, morphological parameters (weight, length, liver weight, carcass weight) were recorded, condition factor (CF) and liver somatic index (LSI) were calculated from these parameters, and a suite of biopsies were collected (as described in Baker et al. [30]). Several biochemical markers (biliary PAH metabolites, EROD activity, liver carboxylesterase, oxidative DNA damage) have been reported in Baker et al. [30]; liver gene expression and liver histology have been reported in Fu et al. [31]; while contaminant levels in the flesh of these fish have been reported in Gagnon et al. [21].

The current study selected a subsample of the fish collected for the above studies to minimize the influence of age and gender as confounding factors [12], which resulted in a low number of fish analysed for each site (*n* = 3–6). In this regard, 22 female fish of similar sizes and ages (between one and two years) were selected (Table 2) (see Baker et al. [30] for age determination methods). Immediately following euthanasia, livers were removed and minced to obtain a homogeneous sample, then subsamples were snap frozen in liquid nitrogen for metabolite (polar and free fatty acids) assessment. This was completed within one minute of euthanasia.

### 4.2. Metabolomics

#### 4.2.1. Polar Metabolites 

Polar metabolites were extracted and detected using methods described in Long et al. [36] and Overgaard et al. [60]. In brief, minced liver (20–30 mg) was extracted in chloroform:methanol:water (1:3:1 *v*/*v*) containing internal standards (140 µM ^13^C_5_-^15^N-Valine and 14 µM ^13^C_6_-Sorbitol) and insoluble material was removed by centrifugation. The supernatant was converted to a two-phase system by the addition of water (final solvent ratio 1:3:3 (*v*:*v*) chloroform:methanol:water) and the upper (aqueous) and lower (organic) phases transferred to separate tubes. A pooled biological quality control (PBQC) was prepared from all the liver extracts and distributed into four 50 µL aliquots. Samples and PBQCs were dried under vacuum with a final wash of 100% methanol and derivatised online prior to analysis by gas chromatography–mass spectrometry (GC-MS). Pooled biological controls were run at regular intervals throughout the batch to check for instrument drift and metabolite stability.

The GC-MS analysis was performed using an Agilent 7890A gas chromatograph coupled to an Agilent 5975C mass spectrometer (Santa Clara, CA, USA) with a Gerstel Autosampler (MPS 2 XL), as described in Long et al. [36]. Chromatograms were imported into Agilent MassHunter Quantitative Analysis software (B.07.0) to perform a targeted profiling analysis as described in Overgaard et al. [60]. Following this, the data were exported as an integrated area matrix for statistical analysis. Although data were collected in an untargeted manner, only metabolites that could be annotated as level 1 identifications (identified compounds, i.e., identifications by chemical reference standards) as prescribed by Sumner et al. [61] were used for statistical analysis.

#### 4.2.2. Free Fatty Acid Metabolites

Free fatty acids were extracted from liver tissue (20–30 mg) using a dedicated method to obtain the quantitative recovery of free fatty acids, rather than using the organic phase from the biphasic extraction described above. Briefly, liver tissue was placed in cryomill tubes containing 200 µL of 100% water-saturated butanol (and 10 µM ^13^C-myristic acid as an internal standard), lysed at 6800 rpm, 3 × 45 s, using a Precellys24 bead-mill attached to a Cryolys cooling unit (Bertin Technologies, France), pre-chilled with liquid nitrogen. The resulting homogenate was transferred to chilled microcentrifuge tubes, and 300 µL of 100% butanol-saturated water was added. Samples were then mixed (Eppendorf, Thermomixer C) for 10 min at 950 rpm at room temperature to maximise free fatty acid extraction, and centrifuged (Beckman Coulter, Microfuge 22R) for 5 min at 15,000 rpm at 0 °C. The supernatant was dried using a rotational vacuum concentrator system (RVC 22-3, John Morris Scientific, Australia), followed by a final wash of 100% methanol to ensure complete dryness. A PBQC was prepared, in the same way as described for polar metabolites, and run at regular intervals throughout the batch.

Samples were derivatised online using 40 µL BSTFA + 1% TMCS (trimethylsilylation) (1 h, 37 °C, 750 rpm) using a Gerstel MPS2 XL autosampler robot (Gerstel, Germany) prior to analysis. One microliter of the derivatised sample was injected in splitless mode using a J&W scientific VF-5 MS capillary column with the dimensions 30 m × 0.25 mm × 0.25 µm (+ 10 m Duraguard phase). Helium at 1 mL/min was used as the carrier gas, with oven conditions set at 35 °C (held for 2 min), then ramped at 15 °C/min to 325 °C (held for 3 min), providing a total run time of about 24 min. Electron ionization (EI) was used to fragment metabolites, and the resultant ions were detected in the range of 50–600 amu. Data processing was performed using Agilent’s MassHunter Quantitative Analysis software, in which target ions for free fatty acids were inspected and integrated as required. A peak area data matrix was exported following the visual inspection and manual integration, where required, of the peaks detected.

### 4.3. Statistical Analysis

#### 4.3.1. Morphological Parameters

Fish standard length and carcass weight (total weight minus viscera and gonads), condition factor (CF), and liver-somatic index (LSI) were compared by analysis of variance (ANOVA) at α = 0.05 after verification of the homogeneity of variance and normal distribution. If data did not conform to the assumptions of ANOVA, they were log_10_ transformed prior to analysis. Tukey’s post-hoc tests were applied to identify inter-site differences at α = 0.05. Data were analysed using Minitab ® (statistical software release 17, Minitab, State College, PA, USA).

#### 4.3.2. Fish Liver Metabolomics

GC-MS peaks were aligned and areas under the peaks integrated. The peak list was pre-treated before statistical analysis to account for biological, experimental, and instrument variations by performing a median normalisation (median of each sample) followed by natural log transformation. A principal component analysis (PCA) and a partial least squares discriminant analysis (PLS-DA) was carried out to investigate the global behaviour of polar and fatty acid metabolites in samples belonging to the different sites. PLS-DA has been used previously to determine separation in metabolite datasets from biological variation [62]. The PLS-DA was run and cross validated, according to Szymańska et al. [63]. Following this, a one-way ANOVA was performed to identify metabolites that changed significantly between the different sites. A follow up post-hoc Fishers LSD test (α = 0.05) was also performed to identify the sites between which the metabolites exhibited significant differences; furthermore, a false discovery rate (FDR) of *p* < 0.05 was applied to control for false positives. An overrepresentation analysis (ORA) was then performed using the polar and free fatty acid metabolites that had been identified as being significantly different. The ORA was conducted between sites where there were greater than three significant metabolites, following the Fishers LSD post hoc test. Statistical analysis of the metabolite datasets was carried out using MetaboAnalyst [64,65].

## 5. Conclusions

This is the first study to use metabolomics to determine the effects of local environmental conditions such as low-level contamination and diet on the sand flathead, an important member of the southern Australian marine fish community. Metabolite profiling of liver tissues could clearly separate fish collected from different sites, which included rural sites with low/no contaminant inputs and urban site recipients of urban and industrial contaminants. Key indicator metabolites included a range of amino acids as well as intermediates in central carbon metabolism and free fatty acid metabolites, reflecting the coverage of multiple interconnected metabolic pathways that underpin the physiological state of these animals. Furthermore, we were also able to attribute these changes to differences in biochemical pathways that are likely to have biological impacts on the fish, mostly changes in energy metabolism. Differences in odd chain free fatty acids showed that metabolomics approaches also have the potential to differentiate fish based on the composition of diet. In this study, we demonstrated that metabolomics approaches can offer a greater level of sensitivity compared to traditional parameters such as morphology and physiological responses with carcass weight, standard length, and CF, most of which showed no or little inter-site differences in the present study. Moreover, the metabolite responses were more informative than traditional biomarkers in terms of biological significance. Further research is needed before metabolomics can be routinely used as a biomonitoring tool; however, this study highlights the potential of using these approaches in wild-caught fish in future biomonitoring programmes.

## Figures and Tables

**Figure 1 metabolites-10-00024-f001:**
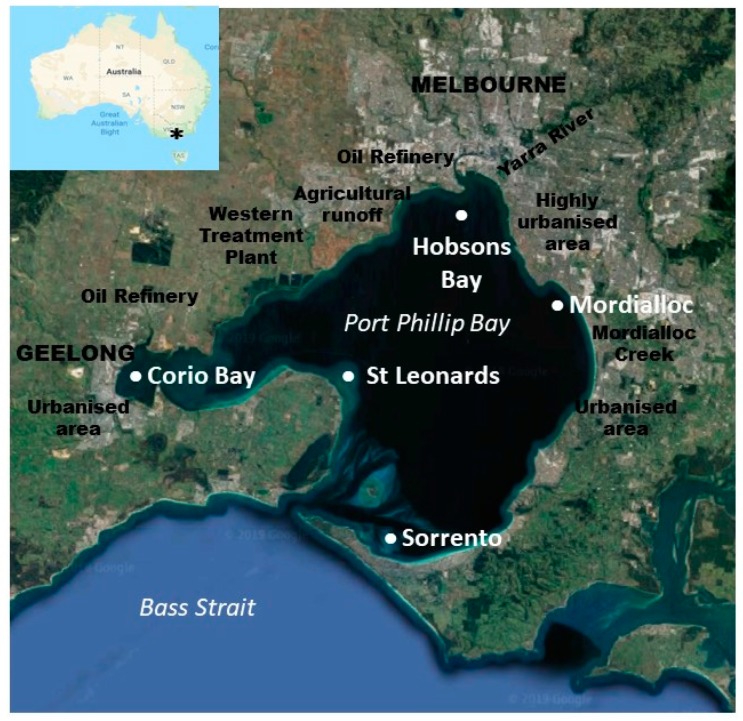
Map showing the location of sampling sites where sand flathead (*Platycephalus bassensis*) were collected in Port Phillip Bay, Victoria, Australia in March 2015. Land use in the surrounding areas and proximity to the cities of Melbourne and Geelong are also included.

**Figure 2 metabolites-10-00024-f002:**
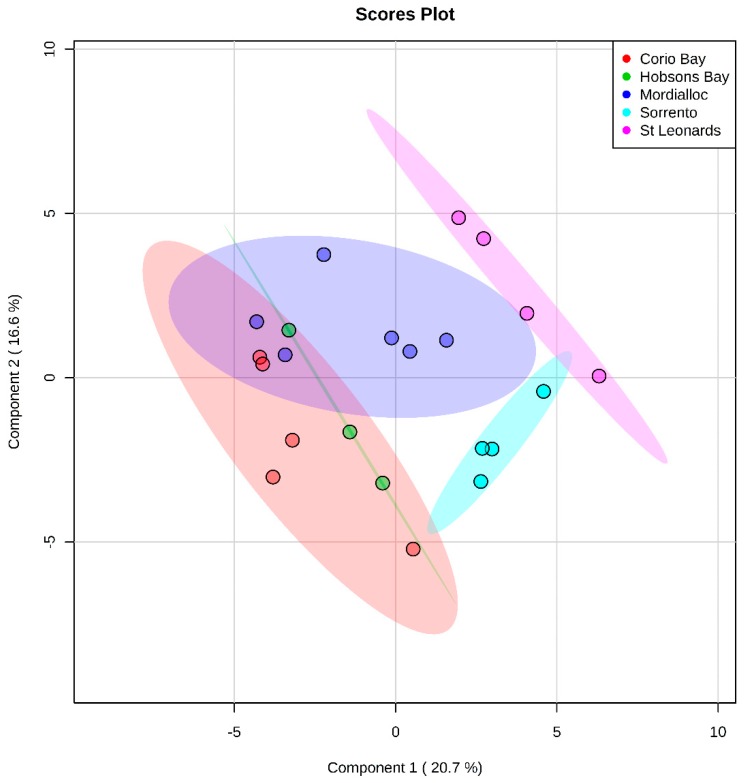
Partial least squares discriminant analysis (PLS-DA) showing the separation of polar metabolites in the livers of female sand flathead (*Platycephalus bassensis*) collected at five sites in Port Phillip Bay, Victoria, Australia in March 2015. Cross validation results are in the Appendix A.

**Figure 3 metabolites-10-00024-f003:**
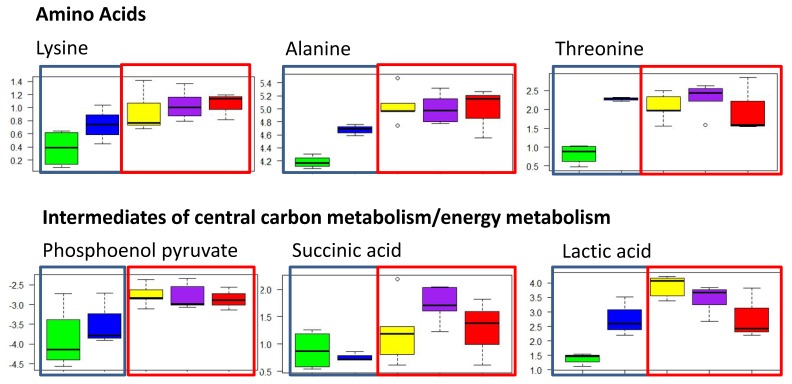
Box plots showing the abundance of different classes of polar metabolites in the livers of female sand flathead collected from five sites (*n* = 3–6 fish per site) around Port Phillip Bay, Victoria, Australia in March 2015. Green = Sorrento, blue = St Leonards, yellow = Corio Bay, purple = Mordialloc, and red = Hobsons Bay. Blue outline represents the rural sites, red outline represents the urban sites.

**Figure 4 metabolites-10-00024-f004:**
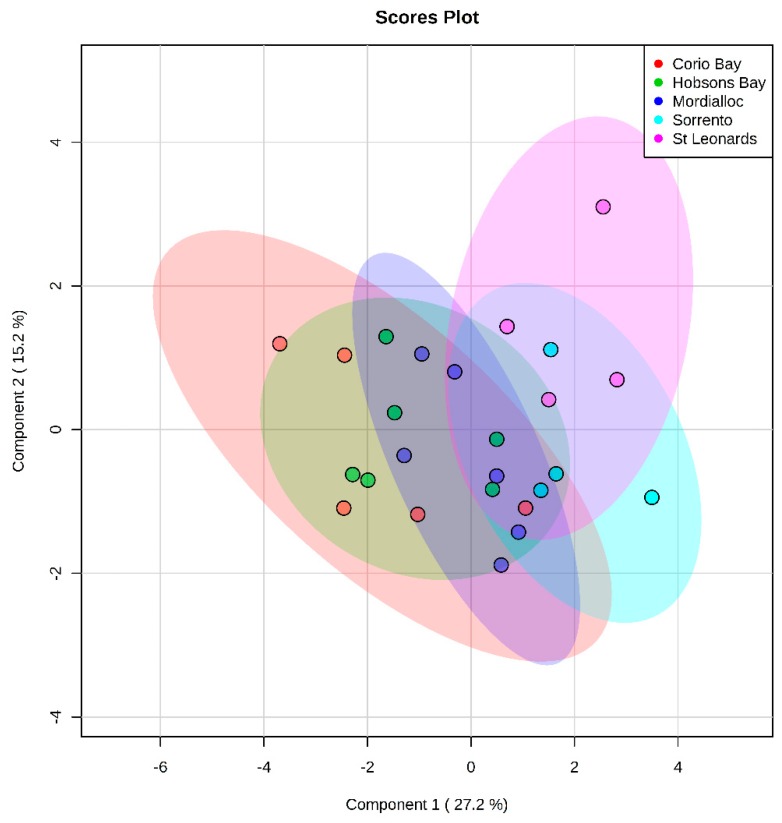
Partial least squares discriminant analysis (PLS-DA) showing the separation of free fatty acid metabolites in the livers of female sand flathead collected at five sites in Port Phillip Bay, Victoria, Australia in March 2015. Cross validation results are in the Appendix A.

**Figure 5 metabolites-10-00024-f005:**
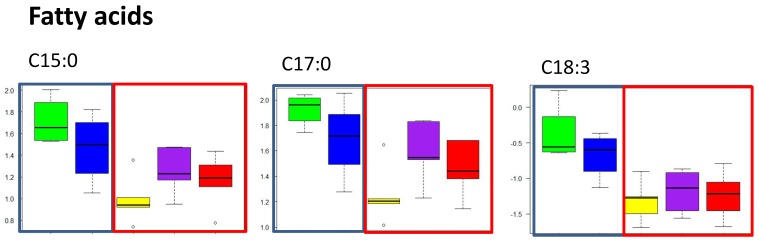
Box plots showing the abundance of some free fatty acid metabolites in the livers of female sand flathead collected from five sites (*n* = 3–6 fish per site) in Port Phillip Bay, Victoria, Australia in March 2015. Green = Sorrento, blue = St Leonards, yellow = Corio Bay, purple = Mordialloc, and red = Hobsons Bay. Blue outline represents rural sites; red outline represents urban sites.

**Table 1 metabolites-10-00024-t001:** Trace elements (mg/kg) found in the flesh of sand flathead collected from Port Phillip Bay in February/March 2015. Results are presented as the mean of the composite pools for each site (± SEM, where possible). Note: Cadmium, chromium, silver, and lead were below the detection limit (0.01, 0.05, 0.02, and 0.01 mg/kg, respectively). No Polycyclic Aromatic Hydrocarbons (PAHs), Organochlorine (OC) pesticides, or Polychlorinated Biphenyls (PCBs) were detectable in any of the samples of sand flathead white muscle collected throughout Port Phillip Bay (adapted from Gagnon et al. [21]).

Site	*N*	As	Cu	Hg	Ni	Se	Zn
Sorrento	4	7.30 ± 0.57	0.16 ± 0.03	0.14 ± 0.02	0.02 ± 0.00	0.45 ± 0.03	5.93 ± 0.85
St Leonards	4	7.87 ± 1.11	0.13 ± 0.01	0.10 ± 0.02	0.02 ± 0.00	0.44 ± 0.04	6.70 ± 0.45
Corio Bay	5	2.38 ± 0.57	0.10 ± 0.01	0.19 ± 0.03	0.01 ± 0.00	0.47 ± 0.02	7.16 ± 0.28
Mordialloc	5	6.40 ± 1.39	0.14 ± 0.00	0.29 ± 0.05	<0.01	0.47 ± 0.02	6.34 ± 0.17
Hobsons Bay	5	2.89 ± 0.75	0.11 ± 0.01	0.20 ± 0.05	<0.01	0.42 ± 0.02	6.46 ± 0.42

**Table 2 metabolites-10-00024-t002:** Mean (+/− SEM) morphometric parameters in female sand flathead (*Platycephalus bassensis*) sampled at different sites within Port Phillip Bay. R = rural site; U = urban site.

Site	Type	*N*	Standard Length (mm)	Carcass Weight (g)	CF ^1^	LSI ^2^
Sorrento	R	4	216.25 ± 3.47^ab^	76.80 ± 4.11^a^	0.76 ± 0.04^a^	0.80 ± 0.04^a^
St. Leonards	R	4	210.75 ± 3.47^ab^	77.35 ± 6.66^a^	0.80 ± 0.04^a^	0.88 ± 0.06^ab^
Corio Bay	U	5	225.40 ± 5.91^a^	98.48 ± 9.21^a^	0.85 ± 0.04^a^	1.21 ± 0.09^cb^
Mordialloc	U	6	200.67 ± 6.41^b^	78.74 ± 8.4^a^	0.87 ± 0.02^a^	1.87 ± 0.23^d^
Hobsons Bay	U	3	226.67 ± 3.71^a^	97.07 ± 6.25^a^	0.83 ± 0.02^a^	1.59 ± 0.09^cd^

^1^ CF calculated as (carcass weight/standard length^3^ (cm))*100. ^2^ LSI calculated as (liver weight/carcass weight)*100. Superscript letters within the table indicate statistical differences (*p* < 0.05) between sites for each variable.

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
