# Peer review of "Metabolomics Provide Sensitive Insights into the Impacts of Low Level Environmental Contamination on Fish Health—A Pilot Study"

_metabolites, 2020, doi:10.3390/metabo10010024_

Round 1

Reviewer 1 Report

The manuscript by Long et al. reports a pilot metabolomics study using field fish samples. Fish were collected five sites of different levels of contamination (three urban, two rural). They detected 16 metabolites, the level of which were significantly different. The sample size was minimal to be able to make conclusions. The authors, however, are careful not to overstate their findings. Manuscript is written logically.

Major comments:
Abstract: I suggest to include a sentence of phrase about metabolomics methodology. When the authors use the word "metabolomics," they may imply only MS metabolomics, although they cited some NMR studies. NMR metabolomics has been extensively applied to ecotoxicological study and thus it would be better to specify the method. In relation to this, the statement "only a few studies have applied metabolomics to investigate... (L50)" is not accurate. Some reviews on NMR metabolomics on aquatic animals are also available.

Cappello et al. Sci Total Environ, 548-549, 13-24, 2016
Cappello et al. Env Poll, 219, 139-148, 2016
Southam et al. J Proteome Res, 7, 5277-5285, 2008

My another concern is that pollutant concentrations of flesh (Table S1) are not very consistent with their urban/rural classification. The As and Ni concentrations are even higher in Sorrento and St. Leonards, which are classified as rural areas. Mordialloc Creek appears to be contaminated according to Table 1, but this data is from 2010-2012. It is not very convincing that the authors attributed the morphological and metabolomic difference to pollution from the data presented. As the authors discussed, most differences can be attributed to the difference in diets. I understand that the authors are careful not to emphasize the effects of pollution, but I feel the tone is still strong. I suggest to change Table S1 to Table 1 and discuss the results more.

L142: The results of PCA are not discussed. Actually, The PCA separated only Sorrento samples from other sites (L140), which is not consistent with the urban/rural classification. How do you interpret these results?

Minor comments:
L65: "and therefore"?
L428: Conclusion is long and not structured well. Can be more concise.

Author Response

Reviewer 1

Comments and Suggestions for Authors

The manuscript by Long et al. reports a pilot metabolomics study using field fish samples. Fish were collected five sites of different levels of contamination (three urban, two rural). They detected 16 metabolites, the level of which were significantly different. The sample size was minimal to be able to make conclusions. The authors, however, are careful not to overstate their findings. Manuscript is written logically.

Major comments:
Abstract: I suggest to include a sentence of phrase about metabolomics methodology. When the authors use the word "metabolomics," they may imply only MS metabolomics, although they cited some NMR studies. NMR metabolomics has been extensively applied to ecotoxicological study and thus it would be better to specify the method. In relation to this, the statement "only a few studies have applied metabolomics to investigate... (L50)" is not accurate. Some reviews on NMR metabolomics on aquatic animals are also available.

Cappello et al. Sci Total Environ, 548-549, 13-24, 2016
Cappello et al. Env Poll, 219, 139-148, 2016
Southam et al. J Proteome Res, 7, 5277-5285, 2008

RESPONSE: Additional clarification on the metabolomics methods used have been added to the abstract, and the wording around the use of metabolomics in aquatic animals has been updated (see line 48-53) and added Cappello et al Env Poll 2016 reference.

My another concern is that pollutant concentrations of flesh (Table S1) are not very consistent with their urban/rural classification. The As and Ni concentrations are even higher in Sorrento and St. Leonards, which are classified as rural areas. Mordialloc Creek appears to be contaminated according to Table 1, but this data is from 2010-2012. It is not very convincing that the authors attributed the morphological and metabolomic difference to pollution from the data presented. As the authors discussed, most differences can be attributed to the difference in diets. I understand that the authors are careful not to emphasize the effects of pollution, but I feel the tone is still strong. I suggest to change Table S1 to Table 1 and discuss the results more.

RESPONSE: Table S1 has been moved to the main manuscript and has replaced the original Table1. Additional information has been added to the introduction to justify the classification of sites used in this study. (see lines 97-114).  Tissue contaminant concentrations as well as a series of physiological, biochemical and histological endpoints have been used to determine this.

L142: The results of PCA are not discussed. Actually, The PCA separated only Sorrento samples from other sites (L140), which is not consistent with the urban/rural classification. How do you interpret these results?

RESPONSE: Following comments from reviewer 2 we carried out a PLS-DA (included in the manuscript (figure 2)), which shows that there is separation between Sorrento and St Leonards and the more urban sites, which is consistent with the ANOVA results and the current discussion

Minor comments:
L65: "and therefore"?

RESPONSE: done.

L428: Conclusion is long and not structured well. Can be more concise.

RESPONSE: The Discussion and Conclusions have been reduced in length

Reviewer 2 Report

Long et al. presented initial evidence of effect of environmental contaminants to fish metabolism. I believe that the paper can be improved from the data analysis and interpretation standpoints and recommend the following -

PCA of fatty acids seems an important data, it should be shown, may be alongside the small metabolite PCA. I would also advise to perform a supervised analysis of the data, such as PLS-DA, and present that in the manuscript. It may be helpful for the readers if the boxplots are color coded by the rural and urban sites, instead of site names. The authors provide two explanations of decreased odd chain fatty acids - line 274 - "a decrease in abundance of these metabolites at urban sites may also be further evidence that fish are using alternative energy pathways to cope with contaminant exposure." and line 299 "We suggest that the significantly lower abundance of odd chain fatty acids in fish from Corio Bay is a result of the lower abundance of phytoplankton at that site. " Which one is more plausible explanation?

Author Response

Reviewer 2 Comments and Suggestions for Authors

Long et al. presented initial evidence of effect of environmental contaminants to fish metabolism. I believe that the paper can be improved from the data analysis and interpretation standpoints and recommend the following -

PCA of fatty acids seems an important data, it should be shown, may be alongside the small metabolite PCA. I would also advise to perform a supervised analysis of the data, such as PLS-DA, and present that in the manuscript.

RESPONSE: PLS-DA were carried out on both the polar and free fatty acid datasets and have now been included in the manuscript (figures 2 and 4), replacing the PCAs. Cross validation results have been included in the supplementary material.

It may be helpful for the readers if the boxplots are color coded by the rural and urban sites, instead of site names.

RESPONSE: We have kept the original colours for the different sites as we wanted to show that there were differences in metabolite abundance between all the sites, but we have included a blue box on all the box plots to represent rural sites and a red box to represent urban sites (figures 3 & 5).

The authors provide two explanations of decreased odd chain fatty acids - line 274 - "a decrease in abundance of these metabolites at urban sites may also be further evidence that fish are using alternative energy pathways to cope with contaminant exposure." and line 299 "We suggest that the significantly lower abundance of odd chain fatty acids in fish from Corio Bay is a result of the lower abundance of phytoplankton at that site. " Which one is more plausible explanation? 

RESPONSE: Further discussion has been added to clarify our view that both contaminant exposure and dietary differences may be contributing to the differences in fatty acid profiles between sites.  We acknowledge that it is beyond the scope of this study to conclude anything specific and highlight the need for further research into this area to better understand how metabolomics approaches can be utilised to unravel the impacts of local conditions on ecological interactions and fish biology.

Reviewer 3 Report

Review report Manuscript ID: metabolites-667614

Overall, I find the paper well written and with novelty linking metabolomics to investigate the effects of pollution on field-collected marine fish. However, some major issues suggest this as a premature study rather than a full publication. The establishment of causality between the contaminants in the sediment and the changes in metabolite profile in the fish is questioned based on several reasons: 1) The authors did not measure the contaminants in the fish target organ (liver), only muscle, this is a large problem. 2) The authors state themselves that in addition to contamination conditions being different, there were differences in primary productivity between the sites. 3) Low number of fish in the study 4) Bioavailability of metal contaminants in the sediment was not addressed. Therefore, could the effects on metabolite profile in fish be due to different nutritional status between the polluted and non-polluted areas? The limitations of biomonitoring, such as confounding factors that are not related to pollution, should be more carefully considered in this work. Overall, conclusion, this is a pilot study and should not be published as is, maybe it could have been part of the studies by Fu et al and Baker et al, but not stand alone.

Minor comments:

Line 51: Not only would diet and temperature affect the metabolite profile of marine organisms, but fish physiology, season, geography would also be major factors contributing to the variability in response parameters.

Line 70: Abbreviation OCs is lacking

Line 119: Table 2, Why did the authors use SEM as a measure of variability, better use standard deviation as this shows the “true” variability. A bit worrying that the number of fish is so low in this study (N=3-6).

Line 126. Although liver index is not a very sensitive marker it serve as an initial screening biomarker to indicate exposure and effects or to provide information on energy reserves/fat. The liver index can be affected by several non-pollutant factors, not only reproductive status, but also disease and nutritional level.

Line 222: there was not enough liver tissue remaining for analysis of contaminants? This is a bit worrying, the concentration of contaminants in the target organ should have been investigated to look for the causative contaminant. The findings from the fillet cannot be used here.

Line 267: The authors have to search the literature, to increase their knowledge. “To our knowledge, there are no studies investigating the effects of chemical exposure on free fatty acid metabolites in fish”

Line 290: I would suggest to use stomach analysis to evaluate the effect of diet and not only measuring the water content of phytoplankton. This is too superficial and can not alone be used as evidence to discuss impact from diet.

Line 295: please change: will have had an impact

Line 355: livers were removed and minced to obtain a homogeneous sample then subsamples were snap frozen in liquid nitrogen for metabolite (polar and free fatty acids) assessment. This procedure is not suitable for metabolite analysis, as the lipids and proteins would start to degenerate during the mincing process. The liver should have been flash frozen immediately upon removal. How much time was used to mince the livers and was the time standardized between the sites and between the fish?

Author Response

Reviewer 3 Comments and Suggestions for Authors

Review report Manuscript ID: metabolites-667614

Overall, I find the paper well written and with novelty linking metabolomics to investigate the effects of pollution on field-collected marine fish. However, some major issues suggest this as a premature study rather than a full publication. The establishment of causality between the contaminants in the sediment and the changes in metabolite profile in the fish is questioned based on several reasons: 1) The authors did not measure the contaminants in the fish target organ (liver), only muscle, this is a large problem. 2) The authors state themselves that in addition to contamination conditions being different, there were differences in primary productivity between the sites. 3) Low number of fish in the study 4) Bioavailability of metal contaminants in the sediment was not addressed. Therefore, could the effects on metabolite profile in fish be due to different nutritional status between the polluted and non-polluted areas? The limitations of biomonitoring, such as confounding factors that are not related to pollution, should be more carefully considered in this work. Overall, conclusion, this is a pilot study and should not be published as is, maybe it could have been part of the studies by Fu et al and Baker et al, but not stand alone.

RESPONSE: As per the comments above, additional information has been added to the introduction to justify the classification of sites used in this study and discussion has been expanded to consider that dietary differences between sites may be contributing to differences in metabolite profiles (lines 104-114). We have also included other confounding factors that could impact the responses, such as geography and season.

Minor comments:

Line 51: Not only would diet and temperature affect the metabolite profile of marine organisms, but fish physiology, season, geography would also be major factors contributing to the variability in response parameters.

RESPONSE: We acknowledge that these other factors may influence variability in response parameters, however given that all sampling was carried out within the same five day period and that the sites used in this study are relatively close to each other and all part of the same large embayment, we don’t think that seasonal or geographical differences between sites would be likely to influence fish responses. The text has been expanded to clarify this point (lines 58-61).

Line 70: Abbreviation OCs is lacking

RESPONSE: Organochlorine pesticides has been added.

Line 119: Table 2, Why did the authors use SEM as a measure of variability, better use standard deviation as this shows the “true” variability. A bit worrying that the number of fish is so low in this study (N=3-6).

RESPONSE: Standard error of the mean is an appropriate measure of variability, and since the focus of the paper was not on these parameters, we don’t think changing them to standard deviation is necessary.

The number of fish used per site was low because we selectively chose to only consider 2-year-old female fish for this exploratory study.  If we had used the entire data set, we would have introduced other confounding factors such as age-based differences in contaminant exposure and previous reproductive activity.  We wanted to remove as many confounding factors as we could, so that we could compare changes in metabolite profiles of ‘like’ fish, where the only real difference was the site that they were collected from.

Line 126. Although liver index is not a very sensitive marker it serve as an initial screening biomarker to indicate exposure and effects or to provide information on energy reserves/fat. The liver index can be affected by several non-pollutant factors, not only reproductive status, but also disease and nutritional level.

RESPONSE: We agree that LSI is not a sensitive indicator but have reported the results for the exact purpose you point out – to be used as an initial screening biomarker.  We acknowledge that the LSI can be influenced by non-pollutant factors as well. 

Line 222: there was not enough liver tissue remaining for analysis of contaminants? This is a bit worrying, the concentration of contaminants in the target organ should have been investigated to look for the causative contaminant. The findings from the fillet cannot be used here.

 RESPONSE: This sentence has been removed.

Line 267: The authors have to search the literature, to increase their knowledge. “To our knowledge, there are no studies investigating the effects of chemical exposure on free fatty acid metabolites in fish”

RESPONSE: This sentence has been removed.

Line 290: I would suggest to use stomach analysis to evaluate the effect of diet and not only measuring the water content of phytoplankton. This is too superficial and can not alone be used as evidence to discuss impact from diet.

RESPONSE: Stomach contents analyses were not available for comparison between sites.  We acknowledge that we cannot make any firm conclusions about the impacts of diet on metabolite profiles on the fish in this study and have updated the discussion accordingly (lines 315-320).

Line 295: please change: will have had an impact

RESPONSE: text changed to “is likely to have influenced” (line 311)

Line 355: livers were removed and minced to obtain a homogeneous sample then subsamples were snap frozen in liquid nitrogen for metabolite (polar and free fatty acids) assessment. This procedure is not suitable for metabolite analysis, as the lipids and proteins would start to degenerate during the mincing process. The liver should have been flash frozen immediately upon removal. How much time was used to mince the livers and was the time standardized between the sites and between the fish?

RESPONSE: We agree that proteins and lipids would start to degenerate very quickly after removal of the liver from the fish. The livers were minced and frozen within one minute of euthanasia. The text has been amended to clarify this point (line 394-395).

Reviewer 4 Report

The manuscript describes a pilot investigation on the health of sand flathead, a marine species caught in the Port Phillip Bay, Australia via liver metabolomics using GC-MS. Five sites within the bay were selected and categorized as rural or urban. Overall, the study is clear and well written. However, there are some concerns which should be addressed.

The authors describe the selection process when choosing which fish to include as part of the current study based on the age and sex of the sand flathead. However, by limiting the variability in the wild-caught fish, the statistical power is also limited. One of the regions, Hobsons Bay, for example, only has an n of 3 fish. Furthermore, two of the five sites only have n of 4 fish. This may be too limited of dataset for a metabolomics study. Line 155: From the author analysis of the metabolite differences, one of the largest differences was between two of the classified “rural” sites “remote from contaminant input”, Sorrento and St. Leonards. Was there a more extensive chemical analysis done in the water to explain differences including organic chemical analyses? Figure 3 and Figure S1: Box plots show relative abundances of different metabolites in fish from the sites. Was there any type of normalization or unit normalization applied to the metabolomic data? There was no information regarding quality control of the analytical methods. The authors should include this information for accuracy and precision of the instrument and stability of the metabolites. In the fish collection and processing, there should be more detail on time between tissue removal and freezing, as well as storage prior to analysis (-80C). Particularly for metabolomics studies, it is crucial to keep tissues frozen, as enzymatic reactions can continue and affect metabolite concentrations.

Specific comments:

Line 148: Table S2 is introduced prior to Table S1.

Figure S1: no S1b label for the free fatty acid metabolites in the figure.

Line 267-268: There are multiple publications on chemical exposure on lipid metabolites in fish. For example, Skelton et al. Comparative biochemistry and physiology 2016, 19, 190-198.

Line 375-377: Mentions that only confident level 1 identifications were used in statistical analysis. Are only those annotated metabolites used to build the PCA plots? I am curious to know how many total metabolites were detected and if those that are not annotated had a significant impact on the overall PCA.

Author Response

Reviewer 4 Comments and Suggestions for Authors

The manuscript describes a pilot investigation on the health of sand flathead, a marine species caught in the Port Phillip Bay, Australia via liver metabolomics using GC-MS. Five sites within the bay were selected and categorized as rural or urban. Overall, the study is clear and well written. However, there are some concerns which should be addressed.

The authors describe the selection process when choosing which fish to include as part of the current study based on the age and sex of the sand flathead. However, by limiting the variability in the wild-caught fish, the statistical power is also limited. One of the regions, Hobsons Bay, for example, only has an n of 3 fish. Furthermore, two of the five sites only have n of 4 fish. This may be too limited of dataset for a metabolomics study.

RESPONSE: As per comments above, the selection of 2-year females only was deliberate to reduce the variability and minimise the influence of confounding factors such as previous contaminant exposure or reproductive activity.  Given this is an exploratory study we feel this was the most appropriate choice, but acknowledge that in future studies, a larger number of fish should be used. 

Line 155: From the author analysis of the metabolite differences, one of the largest differences was between two of the classified “rural” sites “remote from contaminant input”, Sorrento and St. Leonards. Was there a more extensive chemical analysis done in the water to explain differences including organic chemical analyses?

RESPONSE: No, unfortunately there was not any chemical analysis carried out in the water. The sites were classified based on the land use of the surrounding areas.

Figure 3 and Figure S1: Box plots show relative abundances of different metabolites in fish from the sites. Was there any type of normalization or unit normalization applied to the metabolomic data?

RESPONSE: All metabolomics data were median-normalised and natural log transformed prior to statistical analysis, this was described in the Method section.

There was no information regarding quality control of the analytical methods. The authors should include this information for accuracy and precision of the instrument and stability of the metabolites.

RESPONSE: We included details about using pooled biological controls for both polar and free fatty acid metabolites to control for instrument drift and metabolite stability. Additional text has been included for clarity.

In the fish collection and processing, there should be more detail on time between tissue removal and freezing, as well as storage prior to analysis (-80C). Particularly for metabolomics studies, it is crucial to keep tissues frozen, as enzymatic reactions can continue and affect metabolite concentrations.

RESPONSE: We have now included details about the time from removal of the liver to snap freezing in liquid nitrogen.

Specific comments:

Line 148: Table S2 is introduced prior to Table S1.

RESPONSE: This is no longer an issue as we have moved Table S1 to the main body of the manuscript.

Figure S1: no S1b label for the free fatty acid metabolites in the figure.

RESPONSE: A label has been added

Line 267-268: There are multiple publications on chemical exposure on lipid metabolites in fish. For example, Skelton et al. Comparative biochemistry and physiology 2016, 19, 190-198.

RESPONSE: We acknowledge that there are other studies looking at the effect of pollution on lipids, but there are not many specifically related to free fatty acids. We have compared our results to a published paper (Speranza et al).

Line 375-377: Mentions that only confident level 1 identifications were used in statistical analysis. Are only those annotated metabolites used to build the PCA plots? I am curious to know how many total metabolites were detected and if those that are not annotated had a significant impact on the overall PCA.

RESPONSE: We only used level 1 annotated metabolites in the statistical analysis and for the PLS-DA and PCA. We were reluctant to use metabolites that were not annotated in the multivariate analyses as we were specifically interested in what known metabolites were changing between the sites.

Round 2

Reviewer 1 Report

The authors addressed all of my previous concerns.

Reviewer 2 Report

The concerns raised by me in the previous round of review are now sufficiently addressed. The manuscript may be accepted in present form. A thorough check of English language style/grammar may be helpful.

Reviewer 3 Report

Unfortunately, I still think this work is only a pilot study and should not be published as a full research paper. My main concern, which was also highlighted in the first round is the establishment of causality between the contaminants in the sediment and the changes in liver metabolite profile in the fish. Since the authors did not measure the contaminants in the fish target organ (liver), only muscle, this is a large problem. They even did not detect the contaminants in the muscle.. It does not help to hide this information from the paper (line 222), but removing this sentence. Also the low number of fish in the study which where non-randomly selected for the study is problematic. The limitations of biomonitoring, such as confounding factors that are not related to pollution, should be more carefully considered in this work. Overall, conclusion, this is a pilot study and should not be published as is.

Reviewer 4 Report

The authors have addressed my concerns.